# Statistical Experimental Design as a New Approach to Optimize a Solid-State Fermentation Substrate for the Production of Spores and Bioactive Compounds from *Trichoderma asperellum*

**DOI:** 10.3390/jof9111123

**Published:** 2023-11-20

**Authors:** Rayhane Hamrouni, Flor Regus, Magalie Claeys-Bruno, Anne-Marie Farnet Da Silva, Thierry Orsière, Isabelle Laffont-Schwob, Jean-Luc Boudenne, Nathalie Dupuy

**Affiliations:** 1CNRS, IRD, IMBE, Avignon Université, Aix Marseille University, 13013 Marseille, France; flor-bethania.regus@univ-amu.fr (F.R.); m.claeys-bruno@univ-amu.fr (M.C.-B.); anne-marie.farnet@imbe.fr (A.-M.F.D.S.); thierry.orsiere@imbe.fr (T.O.); 2IRD, LPED, UMR 151, Aix Marseille University, 13331 Marseille, France; isabelle.schwob@univ-amu.fr; 3LCE, CNRS, Aix Marseille University, 13331 Marseille, France; jean-luc.boudenne@univ-amu.fr

**Keywords:** solid-state fermentation, *Trichoderma asperellum*, 6-pentyl-alpha-pyrone, lytic enzymes, spores, experimental design

## Abstract

Managing organic agricultural wastes is a challenge in today’s modern agriculture, where the production of different agricultural goods leads to the generation of large amounts of waste, for example, olive pomace and vine shoot in Mediterranean Europe. The discovery of a cost-effective and environment-friendly way to valorize such types of waste in Mediterranean Europe is encouraged by the European Union regulation. As an opportunity, organic agricultural waste could be used as culture media for solid-state fermentation (SSF) for fungal strains. This methodology represents a great opportunity to produce secondary metabolites like 6-pentyl-alpha-pyrone (6-PP), a lactone compound with antifungal properties against phytopathogens, produced by *Trichoderma* spp. Therefore, to reach adequate yields of 6-PP, lytic enzymes, and spores, optimization using specific agricultural cheap local wastes from Southeastern France is in order. The present study was designed to show the applicability of an experimental admixture design to find the optimal formulation that favors the production of 6-PP. To conclude, the optimized formulation of 6-PP production by *Trichoderma* under SSF contains 18% wheat bran, 23% potato flakes, 20% olive pomace, 14% olive oil, 24% oatmeal, and 40% vine shoots.

## 1. Introduction

The valorization of agro-industrial wastes offers many opportunities in terms of waste management. Agricultural wastes are very rich in terms of nutrients, having high carbon content, proteins, fatty acids, and diverse aromas [1,2]. This nutrient richness provides great opportunity for research regarding waste management since agro-industrial wastes can be used as base materials to catalyze the production of diverse compounds [3]. In order to achieve economic significance, the valorization of agriculture wastes should focus on local agriculture products.

One possible valorization of agro-industrial wastes is solid-state fermentation (SSF). SSF is a biotechnological process where microorganisms are cultured in a solid medium in the absence (or near absence) of free water, although some moisture is present to support the microorganism growth and metabolism [4,5]. Many countries in Africa and Europe have published supporting evidence of the use of SSF in agricultural waste valorization, particularly to induce fungal development that can lead to the formation of secondary metabolites, spore production, and the production of enzymes of industrial interest like cellulases, chitinases, amylases, etc. (particularly for fungi) [5,6,7]. However, culture media can have a great impact on fungal development. For example, vine shoots have often been used as structural support for SSF using fungi to produce enzymes such as feruloyl esterases using *Aspergillus terreus* [8,9]. Another example is olive pomace, which, when combined with other agricultural wastes such as winery wastes in SSF, has been found to favor the production of enzymes [10] such as lipases [11]. These types of wastes, vine shoot and olive pomace, are great opportunities for the Mediterranean region since the wine industry produces tons of vine shoots every year that could be used for polyphenol extraction [12,13]. On the other hand, olive oil production in the Mediterranean generates approximately 1 million metric tons of olive pomace [14].

As previously mentioned, SSF has proved to produce secondary metabolites and certain compounds that can be used as antifungal molecules [15,16], offering alternatives for synthetic pesticides. The production of biopesticides has been on the rise [17]: fungi are known to produce many biocontrol compounds through the fermentation process [18]. For example, the genus *Trichoderma* is used as an alternative to synthetic pesticides. *Trichoderma* also has a mycoparasitism mechanism mediated both by the production of lytic enzymes as well as by producing secondary metabolites that inhibit pathogen development [17,18,19,20]. One of the secondary metabolites from SSF using *Trichoderma* is lactone 6-pentyl-alpha-pyrone (6-PP). This molecule is responsible for the coconut-like aroma. Previous studies have demonstrated that 6-PP has antagonistic activities against phytopathogens such as *Botrytis cinerea* [21,22]. Given the right conditions, this compound could be used to protect plants from phytopathogens [23]. SSF for the production of 6-PP is a practical way of producing this compound since the synthetic production is complex and costly [24].

It is important to have in mind that in order to carry out a successful fermentation, some important factors are microorganism growth parameters and the properties of solid support media [25]. Previous studies showed that the composition of culture media plays an important role on the production of 6-PP [26]. Therefore, the optimization of culture media with low cost and local products is a must when considering SSF using *Trichoderma* to produce 6-PP. Media are traditionally optimized by the one-at-a-time strategy, varying one factor while keeping all others constant. Although this strategy is simple and easy to apply without the need for statistical analysis, it involves a large number of experiments, and the interaction among factors is ignored. Because of the importance of admixture design, a study was planned to show the applicability of this method to agriculture wastes and to fit an appropriate admixture regression model, making response variables functions of the proportions of the admixture components.

Finally, the goal of this study was to find an optimum SSF medium using *Trichoderma* that would favor the production of spores, lytic enzymes, and 6-PP. Medium optimization was performed using a surface experimental design to test the effects of five different factors—in this case, five different agricultural wastes. To the authors’ knowledge, this was the first study that intended to valorize local agricultural wastes from Southeastern France.

## 2. Materials and Methods

### 2.1. Fungal Strain and Inoculum Preparation 

The strain of *Trichoderma asperellum* from the collection of team “Biotechnologie Environnementale et Chimiometrie IRD/IMBE Marseille, France” was used for this study. The fungal strain was preserved in cryotubes containing sterile beads (Sigma-Aldrich, St. Louis, MO, USA) and activated in sterilized potato dextrose agar (PDA) culture medium for 5 days at 30 °C.

The inoculum, in the form of spore suspension, was produced by propagating the fungus in an Erlenmeyer flask containing PDA (7 days at 25 °C); this condition has been proven to be the most efficient in the production of spores [6]. The suspension of spores was prepared by adding 0.01% (*v*/*v*) Tween 80 and scraping with a magnetic stirrer to recover the spores. The number of spores was counted using a Malassez cell before inoculating the solid medium. The concentration of the spore suspension used to inoculate the substrate was 2 × 10^7^ spores/g DM.

### 2.2. Substrate Origins and Characteristics 

To ensure a feasible and sustainable final formulation, local agricultural wastes found in the Mediterranean southern region of France were used as substrates for the solid fermentation. Table 1 provides a summary of all the substrates used in this study. Vine shoots were supplied by the Laboratoire Européen d’Extraction (La Laupie, France). Polyphenol extraction was performed through ethanol extraction prior to utilization in solid-state fermentation. Particle sizes ranged from 0.5 to 1 cm. Wheat bran was obtained from a medium-sized industry specializing in the production of baked goods in Marseille, France. Olive pomace was obtained from a three-phase olive mill (Coudoux, France). Olive pomace was dried and grounded (300–1680 μm). Potato flakes, olive oil, and oatmeal were also of French origin and commercially available. Chitin is a commercial product obtained from crab and shrimp shells.

### 2.3. Solid-State Fermentation 

SSF was performed in 250 mL flasks containing 15 g DM (Dry Matter) of substrate admixture. Each admixture was sieved to a 3–4 mm particle size. It is important to mention that the experimental ranges for the five dependent variables of this study represent 50% of the medium. Vine shoots, which acted as a structural support for the fungi, as well as chitin, indeed accounted for 50% of the formulation. For this study, 21 admixtures were prepared according to the experimental plan, as described in Table 2. Each admixture was different in composition. For example, admixture 1 was composed of 20% wheat bran (1.5 g DM), 5% olive pomace (0.38 g DM), 30% oatmeal (2.25 g DM), 30% potato flakes (2.25 g DM), 15% olive oil (1.13 g DM), 40% vine shoots (6.75 g DM), and 10% chitin (0.75 g DM). The humidity of each admixture was adjusted to 50% with distilled water before sterilization. All the culture media were then autoclaved at 121 °C for 30 min. Each admixture was then inoculated with 2 × 10^7^ spores/g DM, the volume of the spore suspension was used to set the final humidity to 66%. The cultures were incubated at 29 °C ± 1 °C for 5 days. Flasks were not hermetically closed to allow some oxygen flow by diffusion [4,5,6,7].

### 2.4. Factors and Domain of Interest

In order to optimize the production of 6-PP, spores, and lytic enzymes, we used design-of-experiments methodology [27]. For that, we first defined experimental ranges for all substrate variables according to the literature [26,28]. The experimental ranges used for this study are presented in Table 3. *x*_1_ wheat bran (varying between 10 and 25%) was selected as a protein and cellulose source, *x*_2_ olive pomace (varying between 5 and 20%) as a lipid source, *x*_3_ oatmeal (varying between 10 and 30%) as a potential protein source, *x*_4_ potato flakes (varying between 10 and 30%) as a starch source, and *x*_5_ olive oil (varying between 0 and 15%) as enzymatic precursor source. It is important to mention that the experimental ranges for the five dependent variables on this study represented 50% of medium. Vine shoots, which acted as a structural support for the fungi, as well as chitin, indeed accounted for 50% of the formulation. 

The SSF responses depended on the proportions of each component in the admixture. Here, the measured responses were assumed to be functionally related only to the proportion of the five components [27,29]. To properly model the responses in all domains of interest, an infinite number of combinations would be necessary; therefore, an empirical mathematical model was postulated using degree 2 admixture model as published by Scheffé et al. [30]:$$Y=\;\beta 1x_{1}+\beta 2x_{2}+\beta 3x_{3}+\beta 4x_{4}+\beta 5x_{5}+\beta 12\;x_{1}x_{2}+\beta 13x_{1}x_{3}+\beta 14x_{1}x_{4}+\beta 15x_{1}x_{5}+\beta 23x_{2}x_{3}+\beta 24x_{2}x_{4}\;+\beta 25x_{2}x_{5}+\beta 34x_{3}x_{4}+\beta 35x_{3}x_{5}+\beta 45x_{4}x_{5}$$

To evaluate the coefficients βi, a set of 21 experiments was selected by using exchange algorithm based on D-optimality criteria presented in Table 2. Condition 21 was followed in triplicate to account for variability.

### 2.5. Validation of Model and of the Optimal Formulation

In order to validate the model, we could have used statistical criteria (R^2^). Nevertheless, we chose, in this study, to confirm the results by comparing the experimental results with those predicted by the model for optimal formulae. For this step, SSF was performed in flask on 15 g (DM) of solid substrate at 29 °C ± 1 in a lab oven, at 66% of initial WHC, and over 5 days. The initial inoculation rate was 2.10^7^ spores/g DM. This experiment was performed in triplicate in order to obtain robust values. 

### 2.6. Desirability

Desirability function, a multicriteria optimization tool, was further investigated for developing a maximum of 6-PP production by *T. asperellum* under SSF in all the experimental domains of interest. This tool was used in order to find the best compromise between the responses: at any point of the domain of interest, predicted responses values were transformed into a desirability function representing the degree of satisfaction. 

When an undesirable value was obtained for one response, the overall desirable value D was 0% and no compromise was found. On the contrary, when each requirement was completely satisfied, the overall desirability value was 100%. Finally, when 0% < D < 100%, an acceptable compromise between the different responses was found. 

### 2.7. Fungal Spore Determination 

The fermented matter (10 g) was added to 100 mL of Tween 80 (0.01%) (Sigma-Aldrich, St. Louis, MO, USA) in an Erlenmeyer flask. A magnetic stirrer was used to release the spores from the solid matter and to homogenize the suspension. Then, 1 mL of the spore suspension was diluted appropriately, and spores were counted using a Mallassez cell (Marienfeld, Lauda-Königshofen, Germany). The results are expressed as spores per gram of dry matter (spores/g DM). 

### 2.8. Enzyme Assays

The fermented material (2g) was placed in a Falcon^®^ tube with 20 mL of distilled water. The enzymatic extract was homogenized in an Ultra-turax (1 min) to obtain a suspension for further determination of enzyme activity. Then, the suspension was centrifuged (5000× *g*, 3 min, 4 °C) and the clear supernatant was used for the assessment of enzymes. The amylase assay was performed according to the method described by Singh et al. [5]. Amylase activity was estimated by analyzing reducing sugar released during hydrolysis of 1% (*w*/*v*) starch in 0.1 M phosphate buffer (pH 6) by enzyme solution incubated at 50 °C for 10 min. Endoglucanase activities were determined using carboxymethyl cellulose (1%) in sodium citrate buffer (50 mM, pH 4.8) at 50 °C for 30 min, according to Nava et al. [31]. Exoglucanase activities were determined using filter paper Whatman No. 1 (1 cm × 5 cm) in sodium citrate buffer (50 mM, pH 4.8) at 50 °C for 60 min according to Singh et al. [32]. The three enzyme determinations mentioned above were stopped using a bath with ice for 5 min. Sugar concentration was determined after each enzyme reaction according to the amount of resulting reducing sugars using 3,5-dinitrosalicylic acid (DNS), and absorbance was measured at 575 nm [32]. Enzymatic activities (U) were defined as 1 μmol of glucose (reduced sugar) released per minute per g substrate dry weight. Finally, lipase activities were determined using 0.5 mL of *p*-nitrophenyl octanoate (25 mM) in phosphate buffer (25 mM, pH 7.0) at 30 °C for 30 min [12]. The calibration curve was performed with *p*-nitrophenol (0–100 ppm) at 412 nm. An enzyme activity (U) was defined as the amount of enzyme required to release 1 μmole of *p*-nitrophenol per minute.

### 2.9. Extraction and Quantification of Volatile Metabolites from Solid Cultures

Extraction of 6-PP was performed by using a soxhlet extraction system from solid fermented material using pure heptane (99.7%, Sigma-Aldrich, St. Louis, MO, USA). For this process, samples (10 g of the fresh fermented material) were co-distilled at 60 °C with 100 mL dichloromethane (99.7%, Sigma-Aldrich) for 2 h. γ-undecanolactone (0.08 mg) (99%, Sigma-Aldrich) was added as an internal standard before extraction. After the extraction, the extract was rinsed from the cell into a bottle and evaporated under reduced pressure at 40 °C. The extract was then filtered with 0.2 μm Millipore filter prior to gas chromatography (GC) analysis. Next, 6-PP analysis was performed on an Agilent Technology gas chromatograph 7890A (GC) equipped with a split injector (split ratio 2:1) operating at 200 °C and a flame ionization detector at 260 °C. Volatile constituents were separated on a Supelcowax capillary column (internal diameter: 0.25 mm, length: 60 m, film thickness: 0.25 μm). The carrier gas was H_2_ (column flow 1 mL/min). The oven temperature was set at 180 °C for 30 min, then raised to 230 °C at 10 °C/min and followed by a final extension at 230 °C for 30 min. For 6-PP analysis, a repeatability analysis was performed; the treatment (extraction and analysis via GC) was repeated 10 times. The generation and the data treatment were performed using Microsoft Excel 2016.

## 3. Results and Discussion

### 3.1. Analysis of Responses 

The model fitting and performance analysis statistical results enabled the discussion of the variation in factors as a function of 6-PP production. Table 3 and Table 4 show the design layout and experimental results for all formulations and for 6-PP production, generated by the design-of-experiments. The data show that the highest 6-PP content for the single component was 19.25 ± 0.3 mg/g DM and the lowest 6-PP content was 0.558 ± 0.7 mg/g DM. Many studies focused on the production of 6-PP using different *Trichoderma* species. Some authors reported a maximum 6-PP yield of 2.54 μg/g DM by *Trichoderma harzianum* 4040 after 7 days of SSF on sugarcane bagasse impregnated to 75% humidity with a nutrient solution [33]. Likewise, our previous study recorded the production of 7.36 ± 0.37 mg/g DM of 6-PP by *Trichoderma asperellum* using an admixture of vine shoots, jatropha cake, olive pomace, and olive oil as a substrate under optimum culture conditions in a bioreactor [17]. On the other hand, Oda et al. [34] obtained higher yields in 6-PP synthesis (7.1 g 6-PP/L) while using a genetically modified *Trichoderma* strain and under liquid-medium conditions. Table 4 also shows that spore concentrations for the 9th run (10% wheat bran, 20% olive pomace, 30% oatmeal, 30% potato flakes, and 10% olive oil) exhibited the highest spore value (5.15 × 10^10^ spores/g DM). In contrast, the 13th run (20% wheat bran, 20% olive pomace, 30% oatmeal, 30% potato flakes, and without olive oil) had the lowest content of spores (5.00 × 10^7^ spores/g DM). This is not surprising since no relationship between 6-PP and spore production has been reported. Research shows that sporulation is not required for 6-PP production [35]. It is explained that the change in the physiological state of *T. harzianum* noticed with the initiation of sporulation results in an important decrease in 6-PP production [36]. Therefore, it seems that an inverse relationship exists between sporulation and 6-PP production.

### 3.2. Modeling of 6-PP Content 

The empirical regression of the model between responses and the five variables for 6-PP contents could be expressed as below: $$y=-1.44x_{1}+20.53x_{2}+5.25x_{3}+1.66x_{4}+1.31x_{5}-37.64x_{1}x_{2}+2.34x_{1}x_{3}+40.04x_{1}x_{4}+41.53x_{1}x_{5}\;-8.32x_{2}x_{3}-14.03x_{2}x_{4}-43.71x_{2}x_{5}+36.17x_{3}x_{4}+21.81x_{3}x_{5}+35.35x_{4}x_{5}$$
where *x*_1_ is wheat bran, *x*_2_ is olive pomace, *x*_3_ is oatmeal, *x*_4_ is potato flakes, and *x*_5_ is olive oil. A positive sign in each equation represents a synergistic effect of the variables. On the other hand, a negative sign represents an antagonistic effect of the variables [37]. From the equation, we identified that the admixtures of wheat bran × oatmeal, wheat bran × potato flakes, wheat bran × olive oil, oatmeal × potato flakes, oatmeal × olive oil, and potato flakes × olive oil show synergic effects. Contrarily, the wheat bran × olive pomace, olive pomace × oatmeal, olive pomace × potato flakes, and olive pomace × olive oil admixtures show antagonistic effects. These results agree well with the arguments by Carboué et al. [4], Hamrouni et al. [6], and González Bautista et al. [38] mentioning that the interaction among ingredients may result in variations in the production of secondary metabolites and spores and lignocellulolytic activities by fungi under SSF. This research is the first step towards a more profound understanding of the interaction between selected ingredients and metabolite production.

### 3.3. Admixture Proportion Optimization

Using the model, we can predict the value of 6-PP production in all the domains of interest. Figure 1 shows four contour plots for 6-PP yield versus different combinations of the five components (wheat bran, olive pomace, oatmeal, potato flakes, olive oil). This experimental design leads to many plots, however. Figure 1 shows the most interesting ones. There is a color gradient where blue represents the lowest concentration of 6-PP (1 mg/g DM) and red corresponds to the highest concentration (15 mg/g DM). The contour lines help define the shaded regions more sharply.

For example, as shown in Figure 1A, when oatmeal was fixed at 0.3 and wheat bran at 0.25, and olive pomace, potato flakes, and olive oil varied in the same way between 0.045 and 0.45, adjusting potato flakes to 0.22, olive pomace to 0.13, and olive oil to 0.05 (point B), we obtained a low amount of 6-PP (0.75 mg/g DM). However, when increasing the proportion of olive oil at 0.15 (Point A; potato flakes 0.17, olive pomace 0.13, and olive oil 0.15), a higher quantity of 6-PP could be found (8.57 mg/g DM).

In Figure 1B, the 6-PP formulation was calculated based on the range of 0.055–0.495 for three factors (proportion of oatmeal, olive pomace, and wheat bran) while olive oil and potato flakes were fixed at 0.15 and 0.3. It can be observed from the comparison of different formulations from the model that the maximum 6-PP value, 8.71 mg/g DM, was achieved by formulation 1 (Point A: 0.28 oatmeal, 0.08 olive pomace, 0.19 wheat bran), followed by 6.89 mg/g DM (Point B: 0.25 oatmeal, 0.17 olive pomace, 0.13 wheat bran) and, lastly, 2.49 mg/g DM for formulation 3 (Point C: 0.16 oatmeal, 0.17 olive pomace, 0.22 wheat bran).

Moreover, in Figure 1C, when adjusting potato flakes at 0.3 and wheat bran at 0.25, and varying oatmeal, olive pomace, and olive oil between 0.045 and 0.405, we obtained the maximum quantity of 6-PP (9.93 mg/g DM, Point A; 0.2 oatmeal, 0.06 olive oil, 0.19 olive pomace). In contrast, when decreasing the proportion of olive pomace and olive oil, there was a lower amount of 6-PP (2.5 mg/g DM, Point B: 0.22 oatmeal, 0.09 olive oil, 0.14 olive pomace).

From the different formulations generated in Figure 1D, to optimally produce 6-PP, the optimum amounts of ingredients selected were as follows: 0.20 oatmeal, 0.15 olive oil, 0.15 wheat bran, 0.20 olive pomace, and 0.30 potato flakes (Point A, 13.47 mg/g DM of 6-PP). Referring to this plot, oatmeal, wheat bran, and olive oil are the main factors that influence the optimization of 6-PP. 

It is evident from this study that the factors (oatmeal, wheat bran, olive pomace, potato flakes, and olive oil) interact with each other and, in turn, affect the responses (6-PP yields) differently. The use of this experimental design allowed the prediction of an optimized culture media that would favor the production of 6-PP. Based on the plot analysis and response (6-PP yields) results and desirability functions, the optimized formulation was determined and corresponds to 18% wheat bran, 23% potato flakes, 20% olive pomace, 24% oatmeal, 14% olive oil, 40% vine shoot, and 10% chitin.

In SSF, the medium is an important factor to account for, being used as both a culture support and substrate. To further explain, as a substrate, it must efficiently provide microbial nutritional needs, and as a support of the culture, it needs favorable physical properties, such as favoring water availability; this substrate should also allow initial conidial anchorage and mycelial elongation in space and mass [39]. The two distinct natures and functional differences between support and substrate are essential and should be used simultaneously, hence requiring the utilization of a combined design. Obviously, when using natural byproducts, the distinction between support and substrate may be blurred: a solid substrate, because of its physical nature, participates as a support in the general texture of the medium [40]. In this study, substrates were selected for their nutritional quality more than for their participation in the general texture of the medium. Moreover, the vine shoots used in this study had previously been submitted to an ethanol extraction to recover the polyphenols they contained and should be considered as constituting an inert ligneous support [28]. It is also interesting because it offers a way of valorization for an overproduced byproduct in Southern France. Concerning substrate effects, 6-PP production is maximized with the presence of 18% wheat bran, 23% potato flakes, 20% olive pomace, 24% oatmeal, 14% olive oil, 40% vine shoot, and 10% chitin. Considering that wheat bran, oatmeal, and potato flakes are richer in carbohydrates and contain starch, it appears that starch is an important compound to maximize 6-PP production. Cost and availability are the main factors considered to select suitable substrates for SSF. The opportunity for agro-industrial byproducts of low commercial value to be used as substrates in SSF is an important economic valorization approach [41]. In the literature, most optimization efforts were made on chemical complementation. The use of pure commercial compounds allows more precise studies to evaluate their effects on metabolism of the fungus and also ensures a high level of repeatability and enhances the process consistency. In fact, using natural byproducts as cultural media may increase the experimental variability between replicates, leading to variations in the process performance because of the complex natures of the substrates [42]. In our study, a good repeatability could be observed because of the inherent homogeneity of the chosen substrates. In order to reduce substrate heterogeneity, it is recommended the byproducts are provided by the same industrial sector have preferentially had the same standardized pretreatment. Of course, the use of a single solid matrix complemented with pure compounds may improve the repeatability of experiments when compared to the use of multiple complex solid substrates. It is, however, possible to use a admixture of solid substrates and to have a good repeatability if the previous recommendations concerning the origin of the medium are respected. This requires having good traceability when it comes to the production chain.

### 3.4. Validation of Model and of the Optimal Formulation

An optimization analysis including twenty-one admixtures was performed using the specific ternary plots model from the Azurad software (v. 10.0). However, only one admixture (18% wheat bran, 20% olive pomace, 24% oatmeal, 23% potato flakes, 14% olive oil), with the highest desirability, was selected for the validation process. The determination of this formula was performed considering only the 6-PP production and enzyme activities since the model for the spore production did not show significant variation. Finally, to confirm the validity of the optimal factors and of the predicted responses calculated, three batches of selected optimized formulations were prepared and analyzed.

The predicted response value of 6-PP was 13.4 mg/g DM, and the experimental value average was 12.64 mg/g DM; this represented 94% of the predicted value (Table 5). The closeness between the predicted and experimental values (evaluated in triplicates for 6-PP) shows there is a consistent repeatability when it comes to this experiment. The difference of 0.76 between the predicted and experimental values confirms that the model has been validated. Thus, these results indicate the success of the admixture design used for the optimization of the substrate formulation for 6-PP production. Under the present culture conditions, we reached concentrations greater than 7.17 × 10^9^ spores/g DM at 5 days of culture, indicating a feasible approach of using agro-industrial wastes as substrates for fungal spore production.

## 4. Conclusions

Experimental designs allowed the modeling and the optimization of 6-PP production. In addition, the equation used represented a more detailed interaction of the response relationship with the five variables and total 6-PP content. Also, the desirability functions defined how the substrate composition could reach the maximum 6-PP content. The optimized medium formulation consists of 18% wheat bran, 20% olive pomace, 24% oatmeal, 23% potato flakes, and 14% olive oil, and this could be used as a substrate in developing biocontrol products in the future. To allow this intention to succeed fully, the safety concerns of the finished products for both human and environmental health (genotoxicity and ecotoxicity) should be evaluated during the development of the formulation. 

## Figures and Tables

**Figure 1 jof-09-01123-f001:**
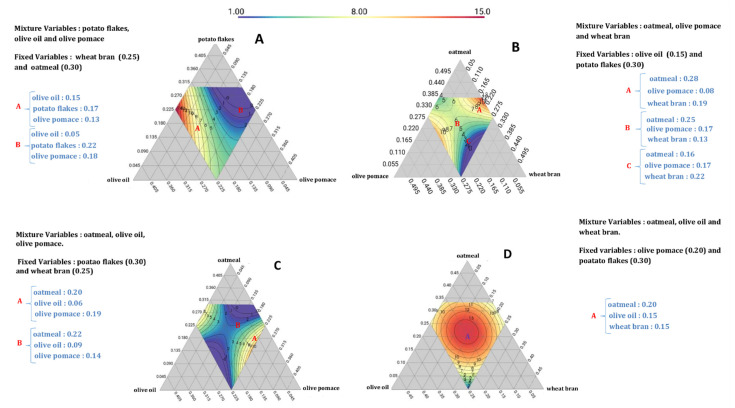
Response surface of the expected 6-PP yield for the experimental approach. (**A**) contour plot; Mixture Variables (potato flakes, olive oil and olive pomace) Fixed Variables (wheat bran and oatmeal), (**B**) contour plot; Mixture Variables (oatmeal, olive pomace and wheat bran) Fixed Variables (olive oil and potato flakes), (**C**) contour plot; Mixture Variables (oatmeal, olive oil, olive pomace), Fixed Variables (poatao flakes and wheat bran), (**D**) contour plot; Mixture Variables (oatmeal, olive oil and wheat bran), Fixed variables (olive pomace and poatato flakes).

**Table 1 jof-09-01123-t001:** Substrate nutritional composition and role in solid-state fermentation [26].

Substrates	Composition (% per 100 g of Dry Matter)	Role in Soil State Fermentation
Vine shoots	Cellulose (51.9), hemicellulose (22.3), lignin (16.6), lipids (0.5), tannins (0.5), ashes (2.5)	Solid support, bringing physical properties like an adapted porosity to the medium
Wheat bran	Holocellulose (58.2), proteins (13.8), lignin (5.7), lipids (7.4), crude fibre (2.1), pectin (1.7), ashes (1.7)	Substrate bringing nutrients to the fungus
Potato four	Carbohydrates (75.2), proteins (9.1), lipids (7.4), crude fibre (2.1), ashes (13.2)	Carbohydrate source
Oatmeal	Proteins (52.4), lipids (8.7), ashes (5.9), Fiber (10.0), Carbohydrates (3.5)	Protein source
Olive cake	lignin (19.5), hemicellulose (16.8), cellulose (11.5), lipids (7.2), proteins (6.5) phenols (1.2)	Oleaginous waste
Olive oil	Palmitic acid (13.6), oleic acid 68.1), linoleic acid (10.2), linolenic acid (0.6)	Lipid source and enzymatic precursors of lipases
Chitin	Natural polysaccharide (β-(1–4)-N-acetyl-d-glucosamine), Aashes (1.4), nitrogen (6.4), lipids (0.6),	Enzymatic precursors of chitinase activities by the fungus

**Table 2 jof-09-01123-t002:** Model conditions followed to optimize spore, enzymatic, and metabolic production of SSF using *Trichoderma asperellum*.

Experience	Wheat Bran (%)	Olive Pomace (%)	Oatmeal (%)	Potato Flakes (%)	Olive Oil (%)
1	20	5	30	30	15
2	25	20	25	30	0
3	25	20	10	30	15
4	25	5	30	25	15
5	25	20	20	20	15
6	10	20	25	30	15
7	10	15	30	30	15
8	25	12.5	17.5	30	15
9	10	20	30	30	10
10	25	10	30	30	5
11	17.5	20	17.5	30	15
12	25	5	25	30	15
13	20	20	30	30	0
14	25	20	30	25	0
15	10	20	30	25	15
16	25	20	17.5	30	7.5
17	18.3	13.3	30	30	8.3
18	25	20	30	10	15
19	25	20	30	17.5	7.5
20	25	12.5	30	17.5	15
21	17.5	20	30	17.5	15

**Table 3 jof-09-01123-t003:** Factors and experimental domain of interest for the solid medium definition.

	Variable	Experimental Domain
Admixture variable	*x*_1_: Wheat bran	10–25%
	*x*_2_: OIive pomace	5–20%
	*x*_3_: Oatmeal	10–30%
	*x*_4_: Potato flakes	10–30%
	*x*_5_: Olive oil	0–15%

**Table 4 jof-09-01123-t004:** Observed responses from the experimental design (the highest result for each of the following parameters is in bold).

Experimental Conditions	Spores (Spores/g DM)	Lytic Enzymes (U/g DM)				
		Amylases	Lipases	Endoglucanases	Exoglucanases	6-PP (mg/g DM)
**1**	7.50 × 10^8^	21.74	8.75	7.07	4.77	12.762
**2**	2.00 × 10^8^	0	1.88	0	0	10.690
**3**	5.00 × 10^8^	36.52	14.55	16.31	8.16	0.558
**4**	5.50 × 10^8^	22.7	13.97	9.80	6.11	**19.250**
**5**	1.10 × 10^9^	8.79	9.75	2.73	1.79	8.730
**6**	5.50 × 10^8^	22.69	17.43	8.06	5.25	5.418
**7**	1.35 × 10^9^	13.36	16.79	4.49	3.45	4.078
**8**	5.50 × 10^8^	4.76	10.41	0.6	0	1.272
**9**	**5.15 × 10** ** ^10^ **	0	7.61	0	0	13.050
**10**	2.20 × 10^9^	7.6	2.47	0	0	0.088
**11**	1.55 × 10^9^	39.3	18.89	**16.99**	8.20	12.178
**12**	6.80 × 10^9^	0	8.73	0	0	3.772
**13**	5.00 × 10^7^	16.33	1.08	0	1.96	2.120
**14**	1.40 × 10^9^	3.16	4.06	0.21	2.31	2.272
**15**	3.05 × 10^9^	19.52	18.20	6.88	3.23	6.658
**16**	1.95 × 10^9^	27.82	6.78	2.02	3.49	7.086
**17**	2.15 × 10^9^	**47.46**	8.41	16.02	**9.97**	5.720
**18**	2.80 × 10^9^	21.34	**19.01**	12.95	5.99	6.032
**19**	1.00 × 10^9^	23.06	11.95	6.85	4.96	2.010
**20**	1.45 × 10^9^	31.86	16.57	9.62	6.7	5.516
**21**	2.45 × 10^9^	3.58	12.98	0	0	14.52
**21″**	3.55 × 10^9^	4.15	13.69	1.26	1.93	6.390
**21″′**	3.45 × 10^9^	3.81	12.71	0	0	7.114

**Table 5 jof-09-01123-t005:** Comparison between experimental and calculated values of responses.

	Calculated Values	Experimental Values
6-PP content (mg/g DM)	13.42	12.64
**Lytic Enzymes (U g^−1^)**		
Amylases activities	22.10	21.46
Lipases activities	13.10	14.67
Endoglucanases activities	5.79	2.19
Exoglucanases activities	3.58	10.93

## Data Availability

Data are contained within the article.

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
