# Peer review of "Statistical Experimental Design as a New Approach to Optimize a Solid-State Fermentation Substrate for the Production of Spores and Bioactive Compounds from Trichoderma asperellum"

_jof, 2023, doi:10.3390/jof9111123_

Round 1

Reviewer 1 Report

Comments and Suggestions for Authors

Dears authors,

I have reviewed the manuscript assigned and I consider that the topic of the manuscript is very interesting because focused on the management of organic agricultural waste which are produced in significant quantities. These wastes could be used to generate bioinputs as alternatives to synthetic pesticides.

However, I make comments and suggestions for the authors that I believe would contribute to improve the manuscript (also see the attached file). 

Related to the methodology described I consider that in some points showld be expanded.

For example:

- In lines 93-98. What was the concentration of the spore  suspension used?

-In lines 112-119. Solid-state fermentation

I consider that the methodology to prepare the dry mass should be expanded.

-Table 1. There are two tables 1. This should be reviewed.

-  Lines 167-190. Fungal spores and enzymatic activities

I consider that the methodology should be expanded.

-Table 3.  The meaning of "E+08 and E+0,9" should be clarified.

Author Response

Dear Reviewer

We greatly appreciate your help in the revision of this manuscript. The comments and all questions have been comprehensive, practical, and instructive. They have helped us improve the paper. Regarding your specific comments, we would like to respond as follows:

  • In lines 93-98. What was the concentration of the spore suspension used?

The concentration of the spore suspension used is 2 × 107 spores /g DM.

We added this information “line 98”: The concentration of the spore suspension used to inoculate the substrate   is 2 × 107 spores /g DM.

  • In lines 112-119. Solid-state fermentation: I consider that the methodology to prepare the dry mass should be expanded.

“line 113” SSF was performed in 250 ml flasks containing 15 g DM of substrate admixture. Each admixture was sieved to a 3–4 mm particle size. It is important to mention that the experimental ranges for the five dependent variables of this study represent 50 % of the medium. Vine shoots, which act as a structural support for the fungi, as well as chitin, indeed account for 50 % of the formulation. For this study 21 admixtures were prepered according to the experimental plan, described in Table 3. Each admixture is different in composition. For example admixture 1 is composed of 20% wheat bran (1.5 g DM) 5% olive pomace (0.38 g DM), 30% oatmeal (2.25 g DM), 30% Potato flakes (2.25 g DM), 15 % olive oil (1.13 g DM), 40% vine shoots (6.75 g DM) and 10% chitin  (0.75 g DM). The humidity of each admixture was adjusted to 50% with distilled water before sterilization. All the culture media were then autoclaved at 121 °C during 30 min. Each admixture was then inoculated with 2×107 spores/gDM, the volume of the spore suspension sets the final humidity to 66%. The cultures were incubated at 29 °C±1 °C for 5 days. Flasks were not hermetically closed to allow some oxygen flow by diffusion.

  • Table 1. There are two tables 1. This should be reviewed.

Modification done : each table is now properly numbered.

4)  Lines 167-190. Fungal spores and enzymatic activities. I consider that the methodology should be expanded.

In order to be as clear as possible and to avoid confusion, this section was revised and divided in 2 parts: 2.7) fungal spores determination and 2.8) enzyme assays.

We added the following information “line171” 2.7 Fungal spore determination:

The fermented matter (10 g) was added to 100 mL of Tween 80 (0.01%) (Sigma-Aldrich, St. Louis, USA) in an Erlenmeyer flask. A magnetic stirrer was used to release the spores from the solid matter and to homogenize the suspension. Then, 1 mL of the spore suspension was diluted appropriately, and spores were counted using a Mallassez cell (Marienfeld, Lauda-Königshofen, Germany). The results are expressed as spores per gram of dry matter (spores/g DM).

We added the following “line178”2.8 Enzyme Assays

The fermented material (2g) was placed in a Falcon® tube with 20 mL of distilled water. The enzymatic extract was homogenized in an Ultra-turax (1 min) to obtain a liquid solid suspension for further determination of enzyme activity. Then, the suspension was centrifuged (5000xg, 3 min, 4°C) and the clear supernatant was used for the assessment of enzymes.

The amylase assay was performed according to the method described by Singh et al. [5]. Amylase activity was estimated by analyzing reducing sugar released during hydrolysis of 1% (w/v) starch in 0.1 M phosphate buffer (pH 6) by enzyme solution incubated at 50 °C for 10 min. Endoglucanase activities were determined using carboxymethyl cellulose (1%) in sodium citrate buffer (50 mM, pH 4.8) at 50 °C for 30 min, in according to Nava et al. [32]. Exoglucanase activities were determined using filter paper Whatman No.1 (1 cm × 5 cm) in sodium citrate buffer (50 mM, pH 4.8) at 50 °C for 60 min according to Singh et al. [33]. The three enzyme determinations mentioned above were stopped using a bath with ice for 5 min. Sugar concentration was determined after each enzyme reaction according to the amount of resulting reducing sugars using 3,5-dinitrosalicylic acid (DNS) and absorbance was measured at 575 nm [33]. Enzymatic activities (U) were defined as 1 μmol of glucose (reduced sugar) released per minute per g substrate dry weight.

Finally, lipases activities were determined using 0.5 mL of p-nitrophenyl octanoate (25 mM) in phosphate buffer (25 mM,pH 7.0) at at 30 °C for 30 min [12]. The calibration curve was done with p-nitrophenol (0–100 ppm) at 412 nm. An enzyme activity (U) was defined as the amount of enzyme required to release 1 μmole of p-nitrophenol per minute.

  • Table 3.  The meaning of "E+08 and E+0,9" should be clarified.

E+08 and E+09 is an alternative way to say 108 and 109, respectively. Modification done in the text

Reviewer 2 Report

Comments and Suggestions for Authors

Review: Statistical experimental design as a new approach to optimize a solid-state fermentation substrate for the production of spores and bioactive compounds from Trichoderma asperellum

The article is nice piece of work, very interesting and dealing with a worldwide problem, which is the valorization of agriculture waste products. In general, the article is well written and structured, however it can be noticeably improved by fixing several issues detected at the materials and method section, and in the results and discussion section. Specific comments are:

Line 94-95: add a reference to support this statement.

Line 103: You are citing Table 1, but there are two tables labelled as number one.

Lines 105-106: it is not clear to which type of agricultural waste you are extracting polyphenols; it must be clarified. Also it is important to add information on why this extraction is necessary.

Line 114: why specifically 66% of moisture content? A reference in support is needed.

Lines 118-119: add a reference in support of this method of fermentation.

Lines 121-126: Section 2.4 Design of experiment; this is information is not a method. Removed it.

Lines 128-154: Section 2.5 Factors and Domain of Interest; This section must be improved. For example, table 2 is cited twice, however it seems you refer to two different tables. The second paragraph repeat information already delivered in the fist one, explanation on the use of Eq. Y must be improved as Coefficients βi are apparently determined in a different experiment, the methods used for such determination also must be explained. Information delivered in lines 148-150 must be mentioned way before in the methodology.

Add a section describing the validation of the model is missing in materials and methods section.

Line 211: Table 4 is missing in the document.

Lines 304-306: This information was not clearly mentioned in the material and methods section. In addition, is this really realistic in case you want to promote this method as cheap alternative to obtain metabolites? Maybe you can commercialize the polyphenols as well. Please elaborate a comment for that reflection.

Line 331: Section 3.4 Validation of model and the optimal formulation is completely missing the Material and Method section (as mentioned before). Include the data for spore and enzyme production with this optimize condition and discuss the result accordingly.

Figure 1: The captions are very small; they are not readable. This must be fixed.

Table 5: This table is not cited in the text.

Author Response

Dear Reviewer

We greatly appreciate your help in the revision of this manuscript. The comments and all questions have been comprehensive, practical, and instructive. They have helped us improve the paper. Regarding your specific comments, we would like to respond as follows:

Review report 2

The article is nice piece of work, very interesting and dealing with a worldwide problem, which is the valorization of agriculture waste products. In general, the article is well written and structured, however it can be noticeably improved by fixing several issues detected at the materials and method section, and in the results and discussion section. Specific comments are:

  • Line 94-95: add a reference to support this statement.

To support this statement, we add the following reference: Hamrouni et al. [6]

We added this reference in the text line95” Hamrouni et al. [6]

  • Line 103: You are citing Table 1, but there are two tables labelled as number one.

Modification done: each table is properly numbered.

  • Lines 105-106: it is not clear to which type of agricultural waste you are extracting polyphenols; it must be clarified. Also it is important to add information on why this extraction is necessary.

The extraction of polyphenols was carried out only for vine shoots.

Wine production is a major agricultural activity in France and produces large amounts of wastes, including important volume of vine shoots. Many studies have shown that a diversity of polyphenol exists in the vine shoot; therefore, it is possible to collect these molecules with interesting properties. This extraction process however does not reduce the volume of the vine shoots, which remains as a lignocellulosic substrate. In the present work, the vine shoots were provided by the Laboratoire Européen d’Extraction (La Laupie, France) after they were previously exhausted of their polyphenol content through ethanol extraction [28]. In this study, vine shoots should not be considered only as a source of nutrients but mainly as a support providing suitable physical textures for the spore anchorage and for mycelial growth of Trichoderma asperellum for the production of 6-PP.

  • Line 114: why specifically 66% of moisture content? A reference in support is needed.

SSF is a process involving microbial growth on the surface and inside a porous matrix in the near-absence of free liquid in the inter-particle volume. In the near-absence of running water in the substrate, water presence depends only on the retention abilities of the substrate namely water holding capacity, WHC (Manpreet et al., 2005). Generally, in SSF the moisture content is between 55% and 75% of WHC. For this study, the quantities of water were chosed based on the water holding capacity of the substrate, corresponding to 66%.

  • Lines 118-119: add a reference in support of this method of fermentation.

The following references were mentioned line « 126 »: De la Cruz-Quiroz [5 ], Maïga [7], Carboué et al[4], Hamrouni et al[6], Carboué et al [28].

  • Lines 121-126: Section 2.4 Design of experiment; this is information is not a method. Removed it.

This section was deleted following the reviewer suggestions.

  • Lines 128-154: Section 2.5 Factors and Domain of Interest; This section must be improved. For example, table 2 is cited twice, however it seems you refer to two different tables. The second paragraph repeat information already delivered in the first one, explanation on the use of Eq. Y must be improved as Coefficients βi are apparently determined in a different experiment, the methods used for such determination also must be explained. Information delivered in lines 148-150 must be mentioned way before in the methodology.

The section 2.5 Factors and Domain of Interest has been revised and the repeated information was deleted. Adittionally, information about the equation Y and Coefficients βi was added “line127” 2.4. Factors and Domain of Interest: in order to optimize the production of 6-PP, spores, and lytic enzymes, we used design of experiments methodology [27]. For that, we first defined experimental ranges for all substrate variables according to literature [26,28]. The experimental ranges used for this study is presented in Table 2. X1 wheat bran (varying between 10 and 25 %) has been selected as a protein and cellulose source, X2 olive pomace (varying between 5 and 20 %) as a lipid source, X3 oatmeal (varying between 10 and 30 %) as a potential protein source, X4 potato flakes (varying between 10 and 30 %) as a starch source and X5 olive oil (varying between 0 and 15 %) as enzymatic precursor source. It is important to mention that the experimental ranges for the five dependent variables on this study represent 50 % of medium. Vine shoots, which act as a structural support for the fungi, as well as chitin, indeed account for 50 % of the formulation.

The solid-state fermentation responses depend on the proportions of each component in the admixture. Here, the measured responses are assumed to be functionally related only to the proportion of the five components [27,29,31]. To properly model the responses in all domain of interest, an infinite number of combinations would be necessary, therefore an empirical mathematical model was postulated using degree 2 admixture model as published by Scheffé et al (1958).

Y

To evaluate the coefficients βi a set of 21 experiments is selected by using exchange algorithm based on D-optimality criteria, presented in Table 3. Admixture number 21 was done in triplicated to account for variability.

  • Add a section describing the validation of the model is missing in materials and methods section.

The following section was added: Validation of model and of the optimal formulation

Line”152” 2.5. Validation of model and of the optimal formulation

In order to validate the model, we can use statistical criteria (R²). Nevertheless, we chose in this study to confirm the results by comparing the experimental results with those predicted by the model for optimal formulae. For this step, SSF will be performed in flask on 15g (DM) of solid substrate, at 29°C in a lab oven, at 66% of initial WHC and during 5 days. The initial inoculation rate will be 2.107 spores/g DM. This experiment will be performed in triplicate in order to obtain robust values.

  • Lines 304-306: This information was not clearly mentioned in the material and methods section. In addition, is this really realistic in case you want to promote this method as cheap alternative to obtain metabolites? Maybe you can commercialize the polyphenols as well. Please elaborate a comment for that reflection.

Thank you for this suggestion, we clarified the materials and methods about the extraction. We have covered this in point 3.

  • Line 331: Section 3.4 Validation of model and the optimal formulation is completely missing the Material and Method section (as mentioned before). Include the data for spore and enzyme production with this optimize condition and discuss the result accordingly.

A section Validation of model and the optimal formulation was added in the materials and methods section. The determination of this formula was done by considering only the 6-PP production and enzyme activities since the model for the spore production didn't show significant variation.

  • Figure 1: The captions are very small; they are not readable. This must be fixed.

This modification was done following the reviewer suggestion.

  • Table 5: This table is not cited in the text.

All tables have been revised and properly cited on the manuscript.